# Significant Microbial Changes Are Evident in the Reproductive Tract of Pregnant Rhesus Monkeys at Mid-Gestation but Their Gut Microbiome Does Not Shift until Late Gestation

**DOI:** 10.3390/microorganisms11061481

**Published:** 2023-06-01

**Authors:** Wellington Z. Amaral, Gabriele R. Lubach, Danielle N. Rendina, Gregory J. Phillips, Mark Lyte, Christopher L. Coe

**Affiliations:** 1UC-Davis Health, Psychiatry and Behavioral Sciences, Sacramento, CA 95817, USA; wzamaral@ucdavis.edu; 2Harlow Center for Biological Psychology, University of Wisconsin, Madison, WI 53715, USA; grlubach@wisc.edu (G.R.L.); danielle.rendina@iff.com (D.N.R.); 3Health and Biosciences, International Flavors & Fragrances (IFF), Wilmington, DE 19803, USA; 4College of Veterinary Medicine, Iowa State University, Ames, IA 50011, USA; gregory@iastate.edu (G.J.P.); mlyte@iastate.edu (M.L.)

**Keywords:** microbiome, firmicutes, *Lactobacillus*, pregnancy, reproduction, vagina, gut, primate, rhesus monkey

## Abstract

**Highlights:**

Bacterial composition and community structure in the lower reproductive tract of female rhesus monkeys are significantly different by mid-pregnancy but significant shifts in their gut microbiome do not occur until later near term.By late pregnancy, four species of *Lactobacillus* and *Bifidobacterium adolescentis* were among the enriched commensals in the hindgut of female monkeys. The abundance of bacterial genes in the intestines, with potential effects on many metabolic pathways, differed in the last month of pregnancy.Several factors likely contributed to some of the differences in the microbial profiles of pregnant monkeys and women, including the consumption of a standardized diet throughout pregnancy and the comparatively low progesterone levels in rhesus monkeys, which were only 2–3% of the third trimester values in women.

**Abstract:**

Vaginal and rectal specimens were obtained from cycling, pregnant, and nursing rhesus monkeys to assess pregnancy-related changes in the commensal bacteria in their reproductive and intestinal tracts. Using 16S rRNA gene amplicon sequencing, significant differences were found only in the vagina at mid-gestation, not in the hindgut. To verify the apparent stability in gut bacterial composition at mid-gestation, the experiment was repeated with additional monkeys, and similar results were found with both 16S rRNA gene amplicon and metagenomic sequencing. A follow-up study investigated if bacterial changes in the hindgut might occur later in pregnancy. Gravid females were assessed closer to term and compared to nonpregnant females. By late pregnancy, significant differences in bacterial composition, including an increased abundance of 4 species of *Lactobacillus* and *Bifidobacterium adolescentis*, were detected, but without a shift in the overall community structure. Progesterone levels were assessed as a possible hormone mediator of bacterial change. The relative abundance of only some taxa (e.g., *Bifidobacteriaceae*) were specifically associated with progesterone. In summary, pregnancy changes the microbial profiles in monkeys, but the bacterial diversity in their lower reproductive tract is different from women, and the composition of their intestinal symbionts remains stable until late gestation when several Firmicutes become more prominent.

## 1. Introduction

For over a century, bacteria in the female reproductive tract, especially lactobacilli, have been thought to be important for reproductive health [1,2,3]. For example, the resident microbial symbionts help reduce susceptibility to sexually transmitted pathogens. In addition, many studies have found that there are also changes in the community structure and a suppression or enrichment of certain taxa during pregnancy and after delivery [4,5,6]. In general, the typical pattern in healthy women is for the *Lactobacillus*-dominated community structure to become more pronounced following conception [7,8]. In contrast, when other taxa are abundant, the resulting dysbiosis can lead to inflammation and a higher likelihood that bacteria can infect placental tissues, which is a major contributor to premature births, along with other factors such as hypertension/preeclampsia [9,10,11].

Research on the intestinal bacteria of pregnant women has indicated that there may also be a directed change in the gut microbiome during pregnancy, helping to facilitate energy harvesting and to promote metabolic adjustments needed by the mother and the fetus [12,13]. Empirical support for this view includes the demonstration that fecal transplants from pregnant women into germ-free mice resulted in weight gains and a decrease in insulin sensitivity reminiscent of late gestation [14]. However, not all studies have been able to confirm the occurrence of large changes in the intestinal bacteria of pregnant women during the first trimester [15,16], with some reporting that the differences became clearer during the third trimester or were more evident in women with large weight gains, especially if maternal obesity was associated with gestational diabetes or hypertension/preeclampsia [17,18,19,20,21]. These findings have led to many experiments in animal models, including both mice and monkeys, that focused on the influence of maternal diet, especially fat content, as a possible mediator of pregnancy-related changes in gut bacteria [22,23]. For example, mice fed a high-fat diet prior to conception were predisposed to larger microbial shifts during pregnancy, further implicating a contribution of food consumption. Thus, one aim of the following research was to determine if conclusions about pregnancy affecting the gut microbiome would generalize to gravid rhesus monkeys consuming the same standardized diet provided to nonpregnant monkeys. All monkeys were fed similar amounts and types of food when cycling, throughout pregnancy, and while nursing after delivery. 

Extensive research has already been conducted on the vaginal microbiota of nonhuman primates, although more typically on cycling, non-pregnant females [24,25]. These studies found that monkeys and apes have more diverse taxa in the lower reproductive tract without the dominance of lactobacilli seen in healthy women. The polymicrobial diversity in nonhuman primates superficially resembles the gynecological condition of bacterial vaginosis, and is associated with a less acidic milieu, lower lactic and glycogen levels, and higher concentrations of proinflammatory cytokines in their cervico-vaginal fluid [26,27,28]. Despite extensive research on cycling primates, pregnancy-related changes in the bacterial composition and community structure of the lower reproductive tract of monkeys have not been investigated as systematically. One paper reported that the normally low levels of lactobacilli in female baboons declined further during pregnancy and the abundance of other taxa increased, including potentially pathogenic bacteria such as *Facklamia* (Family *Aerococcaceae*) [29]. Because this study suggested the monkeys’ vaginal bacterial profiles might have been affected by recent sexual activity, especially if they had multiple male partners, there was still a need to conduct an assessment in a setting where mating activity and husbandry could be controlled. 

Using culture-independent taxonomic profiling with 16S rRNA gene amplicon sequencing, bacterial community structure and taxonomic composition in the reproductive and digestive tracts were first compared at mid-gestation to the bacterial profiles of cycling and nursing monkeys. After finding that the bacterial composition and diversity in the vagina had changed dramatically at mid-gestation but were not different in the hindgut of the pregnant females, the experiment was repeated to verify the fecal bacteria findings with both 16S rRNA gene amplicon and metagenomic sequencing. We then tested the hypothesis that the change in the bacterial composition of the gut might occur later in pregnancy closer to term. In addition, because progesterone had been implicated as a hormone mediator of the increased abundance of bifidobacteria during late pregnancy [30], progesterone levels were also determined in the second experiment. Progesterone levels were assessed both in the blood and fecal specimens, which would be more proximal to the bacteria in the hindgut.

## 2. Material and Methods

### 2.1. Subjects

Specimens were collected from a total of 72 healthy rhesus monkeys (*Macaca mulatta*). These females were members of an established breeding colony that was housed in a large indoor facility with standardized husbandry conditions [31]. All were multiparous adults (mean age 9.5 years, range 5–19 years) and were either not pregnant with regular menstrual cycles, pregnant, or nursing. The females were housed separately from males, either individually, as a pair with another female, or with her infant if nursing. They were all fed the same diet manufactured specifically for primates (Lab Diet 5LFD, St. Louis, MO, USA), approximately 250 g of biscuits daily, and supplemented with a small amount of fruit and vegetables (approximately 150 g, 3–4 times weekly). The light/dark schedule was kept constant year-round at 16 h of light /8 h of dark with lights on at 06:00, a photoperiod that overrides the inherent tendency of rhesus monkeys to reproduce seasonally [32]. This husbandry protocol enabled simultaneous access to nonpregnant females who were cycling, as well as to pregnant and nursing females in each experiment. The schedule for specimen collection in each reproductive phase was predetermined, and all samples were obtained between 09:00 and 11:00. No monkeys were treated with any medications, including antibiotics, and all pregnancies resulted in full-term viable births.

### 2.2. Experimental Design

As shown in Figure 1, Study 1a employed 16S rRNA gene amplicon sequencing to compare the vaginal and gut microbiomes of 10 pregnant monkeys at mid-gestation (3–4.5 months gestational age) to those of 9 cycling and 7 nursing females (2 months postpartum). Study 1b focused on the same 3 reproductive phases with additional monkeys to replicate the 16S rRNA gene amplicon sequencing results from the fecal specimens at mid-gestation and included the additional aim of affirming the Study 1a findings on the enteric microbiome with metagenomic sequencing. Fecal specimens were obtained from 12 more pregnant females and compared to 11 cycling and 11 nursing females (Total N = 34). Microbial DNA was extracted using the same methods as for Study 1a, and the same protocol was used for the 16S rRNA gene amplicon sequencing and data analysis. A second aliquot of the extracted DNA was sent to CosmosID (Germantown, MD, USA) to perform metagenomic sequencing. Finally, a follow-up study was conducted to evaluate fecal specimens collected later in pregnancy (Study 2, see Figure 1). Fecal swabs were obtained from 10 gravid monkeys in the final month of their 5.5-month pregnancy, 3 weeks before delivery. These pregnancy specimens were compared to the fecal bacteria profiles of 12 cycling females, 6 in the follicular and 6 in the luteal phase, so that we could also consider changes in bacterial profiles across the menstrual cycle (i.e., 1–2 weeks after menses or >2 weeks after menses, respectively). In addition to further characterizing gut bacterial changes later in pregnancy, progesterone present in stools was determined, along with quantifying progesterone levels in the systemic circulation. To conduct this hormone assay, small blood samples (2–3 mL) were collected via saphenous venipuncture from all monkeys without sedation on the same morning as fecal specimen collection. For more details on the experimental design and subject numbers in each condition, see Appendix A.

### 2.3. Specimen Collection, DNA Isolation and Sequencing

Vaginal and rectal swabs were obtained with sterile culturettes^TM^ (BBL Culture Swab Collection and Transport System, Becton Dickinson, Cockeysville, MD, USA). Swabs were placed immediately on wet ice and stored frozen in an ultracold freezer at below 70 °C. All specimens from an entire experiment were processed and sequenced as a single batch. The DNA isolation protocol was published previously [33,34], and is detailed in Appendix A. Microbial DNA was isolated in Study 1a and 1b using the PowerSoil DNA isolation kit (MoBio, Carlsbad, CA, USA). DNA samples were purified and quantified with an Invitrogen Qubit 2/4 Fluorometer (Life Technologies, Carlsbad, CA, USA) and dsDNA High Sensitivity Assay Kit (ThermoFisher Scientific, Waltham, MA, USA) and stored at −20 °C in 10 mM Tris buffer until sequenced. DNA samples from Study 1a and 1b were sent to Biosciences Division Environmental Sample Preparation and Sequencing Facility (ESPSF) at the Argonne National Laboratory (Argonne, IL, USA) for sequencing. PCR primers 515f/806r were used to amplify the variable region 4 of the 16S rRNA gene. 16S rRNA libraries were created using Illumina tag PCR reactions with the DNA extracts generated by the Earth Microbiome Project’s protocol [35]. Sequencing was performed using the Illumina MiSeq platform with library preparation according to the Earth Microbiome Project’s protocol. The DNA isolation, library preparation and 16S rRNA gene amplicon sequencing for Studies 1a and 1b were performed with identical methods. In addition, second aliquots of the DNA extracted from the Study 1b fecal specimens afforded the opportunity for deeper metagenomic sequencing by CosmosID and used to affirm the conclusion about the absence of significant change in the gut microbiome at mid-gestation. For this metagenomic sequencing, purified DNA extracts were sequenced on an Illumina HiSeq platform with a paired-end 100 bp single-read configuration. For the subsequent study that focused on late pregnancy, the DNA isolation and 16S rRNA gene amplicon sequencing were conducted with MiSeq v2 chemistry with paired-end 250 bp reads (Wright Labs, Huntingdon, PA, USA). In addition, metagenomic sequencing was employed to enable species identification in the fecal specimens. Libraries were prepared using DNA extracts and the Nextera XT DNA Library Preparation kit (Illumina, San Diego, CA, USA). These libraries were quality-checked, pooled in an equimolar ratio, and sequenced using an Illumina NextSeq to generate 2 × 150 bp reads.

### 2.4. 16S rRNA Gene Amplicon Data Processing

All 16S rRNA gene amplicon data processing and analyses were carried out in-house using identical data pipelines in QIIME2–2022.2 [36,37]. Sequences were filtered by truncating forward and reverse reads to exclude poor-quality sequence reads. QIIME2′s VSEARCH was used to dereplicate and identify sequences by closed-reference clustering and MAFFT-FastTree alignment and phylogenetic tree construction [38,39,40]. The Greengenes (v.13_8) 97%-identity datasets were used as a reference and unmatched sequences were discarded [40]. The feature classifier classify-consensus-vsearch function in QIIME2 was used for taxonomic identification, with reference to the Greengenes database [41,42]. Taxa with low abundance (less than 10 reads across all samples) or low prevalence (present in less than 2 samples) were filtered out to reduce noise in the dataset. Alpha and beta diversity were generated by rarefaction [43] and analyzed in QIIME2. Diversity plots were generated using the ggplot2 package in R [44,45]. Alpha diversity indices (Pielou’s index, Faith’s Phylogenetic Diversity (PD), Observed Species, and Shannon index) were also generated in QIIME2 [46,47,48,49,50]. Beta diversity included weighted and unweighted UniFrac distance matrices and PCoA plots [51,52]. The Kyoto Encyclopedia of Genes and Genomes (KEGG) functional pathways were inferred for the lower reproductive tract in Study 1a using the Phylogenetic Investigation of Communities by Reconstruction of Unobserved States (PICRUSt) tool from the bioBakery suite [53,54]. Additional information on data analyses is provided in Appendix A.

### 2.5. Additional Sequencing in the Second Phase of Study 1

To replicate and extend the findings from Study 1a, on the gut bacterial profile at mid-gestation, fecal specimens were collected from additional monkeys in Study 1b. The methods for the 16S rRNA gene amplicon sequencing were identical to Study 1a. Alpha diversity was calculated similarly to capture the richness and evenness of taxa present in each sample, and beta diversity metrics were used to examine the similarity or dissimilarity of bacterial communities in the hindgut across the 3 reproductive phases. In addition, metagenomic sequencing was used to achieve deeper taxonomic specificity for several a priori comparisons. It was performed in a blinded manner using the CosmosID-HUB Microbiome platform, employing analytic methods described previously for analyzing both human and nonhuman primate samples [55,56,57,58]. In brief, bacterial identifications were provided to the investigator in a tabular format for further analysis, and included taxonomic names, OTU IDs, frequency, and relative abundance. Additional information is available in Appendix A.

### 2.6. Study 2 Analysis of Late Gestation Females

The final experiment employed 16S rRNA gene amplicon and metagenomic sequencing to evaluate the bacterial composition and community structure in fecal specimens from 12 different pregnant females near term. Their gut bacterial profiles were compared to cycling females in both the follicular and luteal phases of the menstrual cycle. DNA from the fecal specimens was extracted using the DNeasy Powersoil Kit (Qiagen, Hilden, Germany). PCR amplification of the 16S rRNA gene and 16S rRNA gene amplicon and metagenomic sequencing were performed by the Wright Labs (Huntingdon, PA, USA). Briefly, 16S rRNA libraries were created using Illumina-tag PCR reactions with the DNA extracts per the Earth Microbiome Project’s protocol [35]. Sequencing was carried out using Illumina MiSeq v2 chemistry with paired-end 250 base pair reads. 16S rRNA data processing was carried out in-house, using the same pipelines employed for Study 1. Taxonomic abundances, indices of alpha diversity, and metrics of beta diversity were analyzed in the same way. Metagenomic libraries were prepared using DNA extracts and the Nextera XT DNA Library Preparation kit (Illumina, San Diego, CA, USA). A total of 20 of the 22 samples yielded >2 million sequences, with a range of 2,979,092–9,536,904 high-quality sequences after data filtration. The data quality was checked, filtered, and annotated using Kraken2 [59]. HUMAnN2 was run on the sequences identified by Kraken2 to annotate sequences with UniRef90 gene annotations, which were regrouped to KEGG orthologs [59,60,61]. The functional capability potentials of bacterial gene profiles in the hindgut of term pregnant females were compared to those of cycling females. For additional details, see Appendix A.

### 2.7. Quantification of Progesterone

Fecal and blood levels of progesterone in the cycling and pregnant monkeys from the second study were determined by mass spectrometry. The LC-MS/MS assay methods have already been published [62,63] and additional details are provided in Appendix A.

### 2.8. Statistical Analyses

Four indices of alpha diversity were analyzed (Pielou’s index, Faith’s Phylogenetic Diversity (PD), Observed Species, and Shannon index) and differences across reproductive phases were compared with Kruskal–Wallis tests. Beta diversity metrics were analyzed using both weighted and unweighted UniFrac distance matrices and PCoA plots [64,65]. The tests of significance included PERMANOVA (if PERMDISP *p* ≥ 0.05), ANOSIM (if PERMDISP *p* ≤ 0.05) and post hoc pairwise comparisons using ANOSIM with 999 permutations. Linear discriminant analysis Effect Size (LEfSe) was used for the multilevel comparison of taxa across reproductive phases using the default statistical thresholds [66,67]. KEGG functional pathways were inferred for the lower reproductive tract (Study 1a) and for the gut (Study 2) using the PICRUSt tool from the bioBakery suite [53,54]. Statistical comparisons of KEGG functional pathways were performed using ANOVA in R, and *p*-values were false discovery rate (FDR)-adjusted. In study 1b, ANOVA was also used to determine if there were significant shifts in the more abundant phyla and genera. The taxa were ranked and relative abundances at mid-gestation were compared to cycling and nursing females. For Study 2, the sPLS-DA tool in the mixOmics package was used in R to identify and visualize relevant metagenomic KEGG orthologs that distinguished the gut metagenomic profiles in late pregnancy [68]. The functional capability potentials of the bacterial genes in the hindgut of the term pregnant females were then compared to those of cycling females. Differences in progesterone between cycling and pregnant females were first compared with one-way ANOVA and then the association between progesterone and bacterial abundance at phylogenetic levels above genus was examined in linear regression models, controlling for the bacterial effects of reproductive phase. The relative abundances of bacteria identified with 16S rRNA gene amplicon sequencing were used to examine associations with serum and fecal progesterone levels after controlling for the potential effect of the reproductive phase. Alpha was set at *p* < 0.01, and *p* values were adjusted for multiple comparisons using the Benjamini–Hochberg method. 

## 3. Results

### 3.1. Study 1

DNA extracted from the paired vaginal and rectal swabs was sequenced for 26 females, either at mid-cycle, 4 months into pregnancy, or 2 months postpartum while nursing. The bacterial community structure and composition in the reproductive tract and hindgut were examined with 16S bacterial rRNA gene amplicon sequencing. The sequencing of the vaginal samples yielded an average 18,587 features, and the rarefaction for the diversity analyses was set at 3977 features per sample to include all monkeys. The sequencing of the fecal specimens averaged 22,388 features, and rarefaction was set at 5986 features per sample. All monkeys were included in the analyses of diversity metrics and taxonomic abundance. 

### 3.2. Reproductive Tract

ANOSIM and PCoA were used to evaluate the similarity or dissimilarity in the community structure of bacteria in the lower reproductive tract and affirmed the significant separation of cycling and pregnant females. Both weighted UNIFRAC analysis, which considers abundance, and unweighted UNIFRAC, which takes the presence and absence of taxa into account, indicated that beta diversity metrics differed significantly across the three reproductive phases (see Appendix A for statistical test values). Post hoc analysis of beta diversity using weighted UNIFRAC values conveyed more specifically that the pregnant females differed significantly from both cycling and nursing females. The PCoA results showing the separate clustering of cycling and pregnant females are illustrated in Figure 2A,B, while also conveying the overlapping patterns for pregnant and nursing females. The unweighted UNIFRAC analysis also indicated that pregnant females differed significantly from cycling females, but not from nursing females (Appendix A for statistical summaries). Over 130 taxa were enriched in the reproductive tract of pregnant females, attaining significant LDA scores > 2.0, *p* < 0.05 (see Appendix A). 

Alpha diversity was evaluated with the Observed Species index, Pielou’s evenness index, Faith’s PD index, and Shannon’s index. All indices of evenness and richness indicated that there were significant differences in vaginal bacteria profiles when the three reproductive phases were compared (Observed Species, H = 13.216, *p* = 0.001; Pielou’s evenness: H = 9.19, *p* = 0.01; Faith’s PD: H = 11.70, *p* = 0.003; Shannon Index: H = 10.825, *p* = 0.004). Post hoc analyses identified the significance as due primarily to the cycling females being different from the two other phases, especially when compared to pregnant females (see Appendix A, for statistical test values). In general, there was more overlap between the pregnancy and nursing phases (see Figure 2C for the illustration of Faith’s PD).

### 3.3. Analysis of Hindgut Bacteria at Mid-Pregnancy

In contrast to the significant changes in the reproductive tract, the alpha diversity and beta diversity metrics of fecal bacteria did not differ significantly across the three reproductive phases. PCoA indicated extensive overlap and similarity in the gut bacterial profiles of pregnant and cycling females (Figure 3A,B). Alpha diversity measures, including Faith’s PD, did not provide any evidence for a directed shift in the richness and evenness of gut bacterial composition at mid-gestation (Figure 3C). LEfSe did identify some differentially expressed taxa in the hindgut, but enriched taxa were more evident in cycling and nursing females, rather than in the pregnant females at mid-gestation (Figure 3D and Appendix A). *Ruminococcus*, which was present at a relative abundance of 1–2% in the hindgut (Table 1), was the only differentially expressed taxa in the hindgut to attain a significant LDA in the pregnant females (illustrated by a small blue vector in Figure 3D).

### 3.4. Comparison of the Reproductive Tract and Hindgut Microbiomes at Mid-Gestation

The 10 bacterial genera that were most prominent in the lower reproductive tract and hindgut are listed in Table 1. *Prevotella* was a major genus present in both vaginal and fecal sequences. In addition, lactobacilli were abundant in both body locations. Although the overall rank order of these taxa remained mostly consistent across the three reproductive phases, there were some significant shifts in relative abundance in the reproductive tract during pregnancy. Specifically, at the phylum level, *Firmicutes* were enriched in the vagina of pregnant monkeys, while *Bacteroidetes* and *Actinobacteria* were higher in cycling females. Further, two bacterial classes, *Clostridia* and *Alphaproteobacteria*, were distinctive in pregnant monkeys, whereas three distinguished the cycling females: *Coriobacteria*, *Bacteroidea*, and *Epsilonproteobacteria*. Among the more abundant bacteria in the vagina, the relative proportions of several taxa within the *Coriobacteriaceae*, *Lachnospiraceae*, and *Ruminococcaceae* families were affected by reproductive phase. It is also noteworthy that the relative abundance of both *Facklamia* and *Streptococcus*, two taxa with potential implications for reproductive health, were higher in the vaginal specimens from pregnant females. 

There also appeared to be some convergence of vaginal and gut microbial profiles in the pregnant and nursing females because of the increased bacterial diversity in the reproductive tract. Differences in the diversity metrics between the two body compartments were most pronounced in the cycling females (F(1,24) = 34.67, *p* < 0.001), and were not significantly different at mid-gestation. Overall, the analyses of taxonomic composition indicated that the nursing females had vaginal microbial profiles that were intermediate between cycling and pregnant females.

### 3.5. Functional Predictions in the Reproductive Tract at Mid-Gestation

Because the female’s reproductive status affected bacterial composition only in the vagina, KEGG analyses were conducted to predict pathway enrichment just for the reproductive tract. Predictive analyses indicated that multiple metabolic pathways might be impacted during pregnancy, potentially influencing several domains of bacterial activity and host physiology. Bacterial genes associated with cellular processes, information processing, metabolism, and host-pathogen interactions were more abundant in pregnant females (Appendix A). For example, there were more bacterial genes in the vagina associated with lipid and amino acid metabolism in pregnant females when compared to the cycling females. The bacterial composition in the vagina during pregnancy could also potentially favor fatty acid and linoleic acid biosynthesis essential for cell membrane production. In addition, the analyses identified pathways associated with amino acid metabolism, including the biosynthesis of lysine, arginine, proline, valine, leucine and isoleucine, as well as a favoring of tyrosine and tryptophan biosynthesis, which might affect the level of bacterially derived neurochemicals in the reproductive tract. Some differences in genes associated with glycan biosynthesis and metabolism were evident in the pregnant females, including fewer genes associated with lipopolysaccharide synthesis, which could potentially reduce proinflammatory signaling to the host. Similarly, there was a reduction in bacterial genes associated with the induction of cellular immune responses. Few differences were found for bacterial genes associated with carbohydrate bio-synthesis and they tended to be less evident at mid-gestation. However, it is important to reiterate that these KEGG predictions reflect bacterial changes in the vagina, not in the digestive tract. 

### 3.6. Study 1b

Specimens from 34 additional monkeys in the same 3 reproductive phases—cycling, mid-pregnancy, or nursing—were assessed with both 16S rRNA gene amplicon sequencing and a metagenomic analysis pipeline to verify if subtle alterations in the gut microbiome might have been missed. The replication experiment also did not detect any significant changes in the gut bacteria at mid-pregnancy when compared to the bacterial community structure and composition of cycling and nursing monkeys. The 16S rRNA gene amplicon sequencing yielded an average of 36,011 distinct features per sample. The rarefaction for the diversity analyses was set to 8793 features to include all samples. In addition to replicating the similarity in alpha and beta diversity metrics at mid-gestation, the LEfSe analysis of bacterial enrichment did not reveal a directed shift in composition at this timepoint in pregnancy.

Even when the second aliquot of the extracted DNA was interrogated more deeply with metagenomic sequencing, there did not appear to be a specific enrichment of any bacterial taxa. *Bacteroidetes* and *Firmicutes* remained dominant in the hindgut during all three reproductive phases. The relative abundances of five phyla are illustrated in Appendix A. The graph also shows that there was not a large increase in *Firmicutes* in the gravid monkeys at this point in mid-pregnancy. *Prevotella* continued to be the most abundant genus detected with metagenomic sequencing, and it was identified more specifically as being primarily *P. copri*. However, neither at the taxonomic level of genus or species was *Prevotella* differentially present at mid-pregnancy (Appendix A). Similarly, the relative abundance of lactobacilli, commensal bacteria with potential benefits for the host, was also not differentially increased in monkeys at mid-pregnancy (Appendix A). Further, the relative abundance of *Bifidobacterium*, a genus reported to increase in pregnant women and some gravid animals in response to rising progesterone levels [30], was not elevated at mid-gestation. The percent of monkeys with *Bifidobacterium* detected at a relative abundance above 0.05% was highest among the nursing females (i.e., at this abundance in 73% of nursing females as compared to only 10% of cycling and 10% of pregnant females). 

### 3.7. Study 2

The sample collection for the second study was scheduled later in pregnancy, three weeks before delivery. Fecal swabs were obtained from 10 late-pregnant monkeys and compared to 12 cycling monkeys, 6 in the follicular and 6 in the luteal phase of the menstrual cycle. The 22 fecal samples were used for both 16S rRNA gene amplicon and metagenomic sequencing.

### 3.8. 16S rRNA Gene Amplicon Findings in Late Pregnancy

The 16S rRNA gene amplicon sequencing yielded an average of 42,605 distinct features per sample. Rarefaction for the diversity analyses was set to 24,400 features to include all samples. Even late in pregnancy near term, this analysis of gut bacteria did not reveal large differences in alpha diversity and beta diversity metrics when compared to cycling females (see Appendix A). The PcoA of beta diversity continued to indicate considerable overlap in the dispersion profiles of cycling and the term pregnant females. However, the sPLS-DA did identify 13 taxa that discriminated between the two menstrual phases. Further, LefSe revealed that 14 taxa were differentially represented in cycling females and that 17 were enriched in pregnant females, all exceeding the cutoff for statistical significance. It is also noteworthy that at least one species of *Lactobacillus* appeared to be enriched in specimens from the term pregnant females, provisionally annotated to be *L. reuteri* (see Appendix A).

### 3.9. Metagenomic Findings in Late Pregnancy

The fecal specimens from cycling and term pregnant females yielded over 2 million high-filtered sequences per monkey for the metagenomic analyses. After data filtration, 2,979,092–9,536,904 sequences were used.

The major taxonomic features distinguishing cycling and pregnant females from this analysis are illustrated in Figure 4. LefSe identified 65 differentially expressed taxa: 59 enriched in cycling females (11 follicular, 48 luteal) and 6 in pregnant females that had significant LDA scores. Specifically, in the term pregnant females, there was a significant enrichment of *Firmicutes* at the phylum level and an increased expression of at least 4 species of *Lactobacillus*, including *L. reuteri*, as well as an enrichment of one species of *Bifidobacterium* (*B. adolescentis*). Conversely, the hindgut of cycling females was enriched with many taxa that could be associated with poor gastrointestinal health, including the genera Campylobacter, species within Helicobacter and Streptococcus, and Proteobacteria at the phylum level (Figure 4B, also see Appendix A for statistical test results).

To explore whether the bacterial profiles in the hindgut could be associated with different functional potentials, PCA and sPLS-DA were employed to assess genomic enrichment. The top 10 bacterial genes that most characterized each reproductive phase are provided in Table 2. The PCA did not identify a global shift in bacterial genes across reproductive phases. However, it indicated that there was a more homogeneous clustering of bacterial genes in the hindgut of the late-pregnant females and a more divergent gene dispersion during the follicular phase (illustrated by smaller and larger ellipses, respectively, in Figure 5A). The sPLS-DA also identified subsets of bacterial genes that distinguished the late-pregnant females from cycling monkeys in the luteal phase (sPLS-DA Component 1; Figure 5B) and in the follicular phase (sPLS-DA Component 2; Figure 5B). Component 1 was comprised of 300 bacterial genes from both cycling and pregnant monkeys; Component 2 consisted of 10 genes that were specifically enriched in specimens collected during the follicular phase (see Appendix A for a complete listing of the 310 genes). The effect sizes for the bacterial genes in each component, 300 and 10 genes, respectively, are also provided in Appendix A. Many of these bacterial genes present in the hindgut of the gravid monkey might influence intestinal physiology or maternal metabolism and thereby indirectly affect fetal development. For example, they could potentially influence maternal weight gain in late pregnancy because several of the gene function pathways were associated with lipid metabolism. Others could have direct effects. For example, there was a differential abundance of genes that regulate the producton of aminocacytl-tRNA synthetases, which are essential for protein synthesis. In addition, the differences included bacterial genes that encode RecQ DNA helicases, which are essential for genome stability and the repair of replicative damage. In total, 45 gene pathways were found to be significantly enriched in pregnant females, and 19 were enriched in cycling females.

### 3.10. Progesterone Levels in Cycling and Late-Pregnant Females

Progesterone levels in systemic circulation as well as present in fecal samples were compared in the cycling and late-pregnant females 3 weeks before delivery. Blood progesterone levels were higher in the luteal than follicular phase and were further elevated in the pregnant monkeys, although still reaching only a mean level of 3.2 ng/mL in the final month of pregnancy (F[2,19] = 5.09, *p* = 0.017) (Table 3). Substantial amounts of progesterone were also excreted via the hindgut in the ng/g range, with a significant increment in fecal progesterone levels during late pregnancy (F[2,19] = 4.94, *p* = 0.019). However, there was not a clear influence of either blood or fecal progesterone levels on any of the bacterial diversity metrics. In addition, there were only a few significant associations evident between fecal progesterone and bacterial abundance.

Specifically, only the relationships between fecal progesterone and the abundance of bacteria within the *Bifdobacteriacaeae* and *Pasteurellaceae* families attained statistical significance when considering the differences across reproductive phases. The relative abundance of the *Bifidobacteriaceae* was most strongly associated with the fecal progesterone levels (β = 10.474, F = 48.129, *p* < 0.001, adj. *p* < 0.001). Within *Pasteurellaceae*, two genera exhibited significant positive associations with fecal progesterone levels: *Aggregatibacter* (β = 12.746, F = 34.975, *p* < 0.001, adj. *p* < 0.001) and *Haemophilus* (β = 1.558, F = 18.546, *p* < 0.001, adj. *p* = 0.013). No associations were found between serum progesterone and any bacterial taxa. The likely involvement of progesterone as the primary hormone mediator of bacterial change in late pregnancy may also be limited by the low correlation between blood and fecal hormone levels. When considering just the 10 paired blood/fecal values in pregnant females, the correlation was small (r = 0.11, NS). The association trended only moderately higher when considering all paired values in both cycling and pregnant females (r = 0.32, NS).

## 4. Discussion

Our findings on bacterial changes in the reproductive and digestive tracts of pregnant monkeys concur with many previous papers reporting significant shifts during pregnancy, both in women and other animal species [69,70,71,72]. However, the vaginal alterations in pregnant rhesus monkeys did not involve the increased dominance of lactobacilli that is usually observed in healthy women, but rather a further enhancement of the microbial diversity already evident in the cycling female monkeys prior to conception [24,34,73]. KEGG analyses of the bacterial genes that would accompany the increased bacterial diversity suggested that differences in taxonomic composition could potentially impact several metabolic pathways in the reproductive tract, as well as influence protein signaling to the placenta and contribute to the immunomodulation needed to prevent maternal rejection of the fetus. It is also noteworthy that we detected a pregnancy-related increase in the relative abundance of *Facklamia* in the vaginal specimens—this Gram-positive anaerobe had been reported to be higher in the lower reproductive tract of pregnant baboons under natural conditions [29]. It can be a pathogen in women and has been implicated as a bacterial cause of urinary tract infections and chorioamnionitis during pregnancy [74]. *Facklamia* was even more common in the vaginal specimens collected at 2 months postpartum, but in the pregnant monkeys, it was accompanied by an increase in the relative abundance of *Streptococcus*, a genus that includes some pathogenic species that can be a concern for maternal health and neonates during delivery [75]. Despite the presence of potential pathogens in the vaginal specimens, these bacteria do not appear to be overtly harmful to female monkeys nor did they affect pregnancy outcomes. In fact, the reproductive success of the rhesus monkeys in this breeding colony is very high when compared to fertility and fecundity rates in other programs and to primates living under natural conditions [32]. None of the pregnant females assessed in our experiments experienced miscarriages or had premature births. We did not observe the associations between bacterial pathogens, chorioamnionitis, and preterm birth that have been reported in clinical studies in humans [76].

It is also important to reiterate that the extent of the bacterial diversity in the vaginal specimens is congruent with previous evaluations of other primate species. Monkeys typically have a median of 10 different taxa in the lower genital tract (with a range of 6–19 genera) [24]. The presence of these bacteria is not typically associated with adverse symptoms, despite the presence of substantial amounts of sialidase enzyme activity in their cervico-vaginal fluid, which is a common clinical marker of bacterial vaginosis in women [34]. The presence of diverse taxa in the vagina of monkeys, including many that are more typically associated with the intestines, may occur because monkeys are quadrupedal. Their vaginal opening is located below the anus, which can increase the exposure to fecal material. This anatomical position has been shown previously to facilitate bacterial transfer in quadrupedal farm animals [77].

The major taxa we detected in the hindgut of female rhesus monkeys were also in keeping with previous research on primates. *Prevotella* was the most abundant genus in fecal specimens; its prevalence exerts a strong influence on the overall community structure of the gut bacteria in monkeys. Thus, the continuation of high levels of *Prevotella* during pregnancy may account for the stability of the bacterial composition in the hindgut through mid-gestation. It is noteworthy that there was more similarity in the bacterial diversity in the gut and reproductive tract at this point in pregnancy, a convergence that has also been observed in some pregnant women [78]. Only when the sample collection was scheduled later in the final month of pregnancy were significant increases in the relative abundance of Firmicutes detected. The lateness of this shift would concur with the delayed and/or larger shift in bacterial profiles that many investigators have reported for healthy pregnant women during the third trimester.

The use of metagenomic sequencing in Study 2 enabled us to identify several enriched species of *Lactobacillus*, including *L. reuteri*, and one species of *Bifidobacterium* (*B. adolescentis*). The enrichment of *L. reuteri* is of particular interest because these bacteria synthesize a soluble peptide or metabolite that can stimulate the release of oxytocin, which could act on uterine and mammary tissue in a parturient female and/or facilitate maternal responsiveness in an expectant mother [79]. However, unlike some previous evaluations of pregnant women during the third trimester, *Proteobacteria* were not enriched in the term pregnant monkeys and were more abundant in the cycling females. In addition, many intestinal bacteria with a potential for pathogenic actions were higher in cycling females, including species within the Order *Campylobacterales* as well as the *Helicobacteraceae* and *Streptococcaceae* families, which may be indicative of a suppressive regulatory process that helps to promote a healthier microbial balance during late pregnancy [80].

The high proportion of *Firmicutes* and *Bacteroides* in the hindgut of rhesus monkeys during all reproductive phases is reminiscent of the intestinal microbiomes described for humans eating a high fiber diet, rather than when consuming the high fat diet common in Western countries [81,82]. However, it is important not to over-generalize from the current findings on rhesus monkeys to all primates because many different gut microbial phenotypes have already been described in the >600 species of nonhuman primates [83,84]. Studies conducted in their natural habitats have documented extensive differences in gut microbiomes related to the dietary specialization of each species [85,86]. For example, the gut microbiota of monkeys with a leaf diet is very different than the commensals found in the more omnivorous rhesus monkey, especially when consuming the usual diets provided in laboratory and zoo settings [87]. Moreover, even the typical gut microbiome of a monkey can be experimentally modified by increasing the fat or fiber content of the food [45]. Of relevance to the current analysis of pregnancy-related changes, it has been shown that a high-fat diet can impact the gut microbiomes of pregnant monkeys, which in turn can influence the bacteria transferred from the parturient female to her infant after birth [88,89]. Further, the gut microbiome of pregnant monkeys can be differentially affected by whether they become obese when consuming a high fat diet.

Based on the previously described effects of diet, both the food constituents and portion sizes were controlled in the current research so that the type of food being consumed would be similar in all reproductive phases. Our husbandry practices also employed an invariant photoperiod with 16 h of light and 8 h of dark year-round to override the natural tendency of rhesus monkeys to breed seasonally [32]. This protocol permitted the contemporaneous collection of specimens from cycling, pregnant and nursing females in each experiment. Otherwise, samples from each reproductive phase would have to be collected at different times of the year. If not controlled, the monkey’s reproductive behavior would follow an annual cycle. Under natural conditions, it would thus be accompanied by seasonal variations in food abundance and types of fruit, seeds and leaves available. Even in humans who still adhere to a traditional hunter-gatherer lifestyle, it is known that seasonal variations in their diet lead to a corresponding annual cycle in the composition of their gut bacteria [89].

Although the gut bacteria of pregnant rhesus monkeys did differ near term, it is important to reiterate that the change was delayed and not overtly evident at mid-gestation. Similarly, not all studies on pregnant women have found significant changes in gut bacteria early in the first trimester [69,70,71,72]. Moreover, the later shift in the third trimester is typically described as being larger. The delay in pregnant monkeys could also reflect that their gestational weight gain is comparatively small, typically just 2–3 kg [90]. Thus, it may not be as essential to rapidly increase lipogenesis and glucogenesis to the extent required in pregnant women. In rhesus monkeys, it may be sufficient for the larger changes in maternal metabolism and gut bacteria to occur when fetal growth is maximal during the final month of gestation. However, even if evinced only in late pregnancy, potentiating maternal weight gain is still important in monkeys because the amount of weight gained by the dam has a significant effect on the birthweight of infants at delivery [91]. Heavier pregnant monkeys that gain more weight typically give birth to larger infants.

Our studies also identified one hormonal factor that may account for the delay in gut bacterial change in pregnant rhesus monkeys. In contrast to the large increase in progesterone that occurs in pregnant women, which is already evident in mid-pregnancy and then rises to over 150–200 ng/mL in the third trimester [92], the progesterone increase was surprisingly small in gravid rhesus monkeys. At 3 weeks before delivery, the mean progesterone level in the blood was only 3.2 ng/mL. Given that the placenta is the primary source of progesterone during pregnancy, and the structure of the hemochorial placenta is very similar in monkeys and humans, the extent of this species difference in endocrine physiology was unexpected. Progesterone in pregnant monkeys is not commonly studied today, but publications from over 4 decades ago have reported similarly low levels of progesterone in pregnant rhesus monkeys and the closely related cynomolgus monkey [93,94]. By also measuring progesterone present in fecal specimens, we confirmed that bacteria in the hindgut would still be exposed to excreted progesterone. However, we could not show that fecal progesterone was the hormone mediator of most bacterial changes in the gut. Higher levels were correlated with the enrichment of taxa in only two bacterial families, *Bifdobacteriacaeae* and *Pasteurellaceae*, during late pregnancy.

Several limitations of our studies should be acknowledged. While the total number of monkeys evaluated was large for primate research, the number of pregnant monkeys in each experiment precluded further categorization into additional subgroups. For example, the number of pregnant females was not sufficiently large to be powered enough to divide them into different weight categories. For that reason, we could not consider the potential influence of maternal adiposity and gestational weight gain. Both are known to be important for understanding the gut microbiomes of pregnant women and animals [95,96,97,98,99,100,101]. In addition, one of our selection criteria for pregnant monkeys was that the female should be multiparous. Thus, it was not possible to compare differences between first pregnancies and subsequent pregnancies in older females. Studies in other animal models, including pregnant pigs, have documented larger gut bacterial changes with increasing parity [102]. In addition, the gut bacterial changes may begin sooner after conception in more parous females. Our experiments were also purposefully designed to be cross-sectional so that the microbial profiles of nonpregnant and pregnant females could be compared simultaneously with all specimens sequenced together as a single batch. Therefore, it will be important to verify our conclusions with a prospective collection of serial specimens from the same female throughout pregnancy. Finally, we should acknowledge that different sequencing methods and analytical tools are known to yield divergent results and conclusions about relative abundance, taxonomic diversity, and community structure [103,104,105,106]. Identification of bacterial species is often more accurate with metagenomic sequencing, and it provides a more precise analytical platform for predicting gene function pathways.

## 5. Conclusions

Notwithstanding these caveats, which can be addressed in future studies, the findings do advance our comparative knowledge of nonhuman primates and highlight their value as a model for investigating many clinical conditions [107]. Significant differences in the taxonomic composition of bacteria in the vagina were observed by mid-gestation, and the shift involved an increase in bacterial diversity, rather than the more pronounced dominance of *Lactobacillus* seen in healthy, pregnant women. Although this bacterial diversity is the normal condition for nonhuman primates, it is possible to experimentally induce higher levels of lactobacilli in their lower reproductive tract by creating a more permissive environment with prebiotic glycogen substrates or with probiotic lactobacilli suppositories [108,109]. Another novel observation was the delay in the bacterial shift in the hindgut until late pregnancy. The lateness could have been due in part to the provision of a standardized diet, including a fixed portion size, which would have prevented extreme weight gains. For example, we did not detect increases in Enterobacteriaceae and Escherichia coli, which have been described in some pregnant women and animal species with large gestational weight gains [15]. In addition, none of the gravid monkeys experienced any of the clinical complications associated with obesity, including gestational diabetes or preeclampsia, which have been associated with altered gut microbiomes in pregnant women [71].

By late pregnancy, there was an increase in *Firmicutes*, including several species of *Lactobacillus*, and an enrichment of *B. adolescentis*, a taxon of relatively low abundance. In addition, the composition of the gut bacteria was associated with a distinct clustering of 310 genes, which could affect myriad functional pathways. Some could facilitate a remodeling of the maternal intestine or influence digestive physiology during pregnancy. Many were associated with lipid metabolism, glycolysis and gluconeogenic metabolic pathways, which are important for maternal health [110,111]. Even without bacterial translocation, the release and transfer of soluble factors from bacteria into circulation, including short-chain fatty acids and serotonergic metabolites, have been found to be associated with gestational hypertension and diabetes [112,113]. Several unique bacterial genes were also detected that warrant further investigation. For example, we identified a pregnancy-related increase in the abundance of gut bacterial genes encoding aminoacyl-tRNA synthetases, which regulate enzymes essential for protein synthesis [114]. In addition, there was an increased abundance of bacterial genes in the hindgut that encode RecQ helicases, which are required for the repair of replicative damage and known to be conserved from bacteria to mammals [115]. Beyond the possible effects on maternal physiology during pregnancy, bacteria in both the gut and reproductive tract help to seed a microbial endowment to mammary tissue and breast milk and will become established as the microbiome of the neonate and developing infant [116,117]. Finally, there are some implications for clinical practice and policy because the gut microbial profile during pregnancy is known to be responsive to dietary supplements [118], and the extent of the transfer of beneficial commensals from mother to infant is amenable to obstetric decisions during delivery [119,120].

## Figures and Tables

**Figure 1 microorganisms-11-01481-f001:**
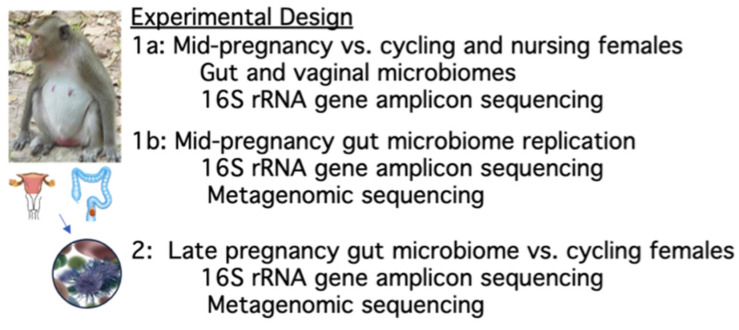
Experimental plan for the two studies. Study 1 was conducted in two phases and investigated whether bacterial profiles in the vagina and hindgut were different at mid-pregnancy. The second study evaluated the gut microbiome later in pregnancy near term. All samples were analyzed with 16S rRNA gene amplicon sequencing. Metagenomic sequencing was also employed to analyze the fecal specimens in Studies 1b and 2.

**Figure 2 microorganisms-11-01481-f002:**
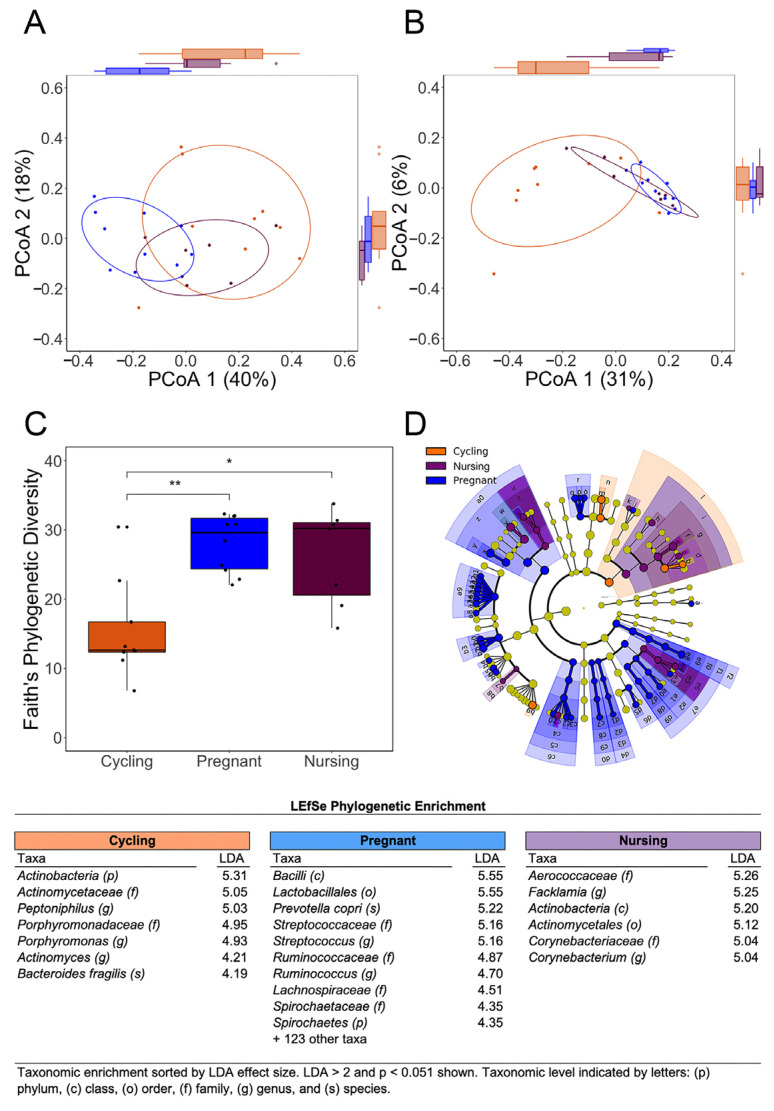
Vaginal microbiome of rhesus monkeys at mid-gestation compared to cycling and nursing females. The PCoA of weighted and unweighted UNIFRAC distances indicated a significant separation of females by reproductive phase ((**A**), weighted, and (**B**), unweighted). Alpha diversity indices, including Faith’s PD, were significantly higher during pregnancy and nursing when compared to cycling females (**C**). Significance indicated by asterisks (* *p* = 0.05, ** *p* = 0.01). The cladogram illustrates distinctive taxonomic features in each phase by color and height of the vectors (**D**). Colored circles within cladogram connote taxa at different phylogenetic levels that delineated each reproductive phase (from species to kingdom level, innermost to outer rings, respectively). LEfSe indicated that the relative abundances of many taxa were significantly different during the 3 phases, with 133 higher in pregnant females.

**Figure 3 microorganisms-11-01481-f003:**
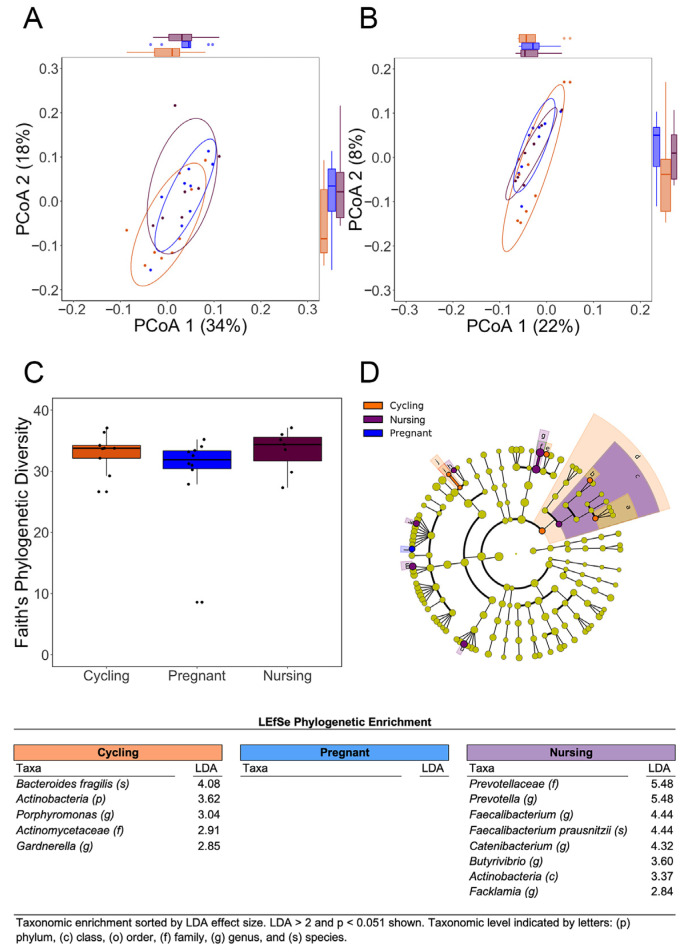
The gut bacterial profiles of pregnant monkeys at mid-gestation did not differ significantly from cycling and nursing females, as shown by the overlapping clusters in the PCoA of UNIFRAC distances ((**A**), weighted, and (**B**), unweighted). In addition, there was not a significant change in alpha diversity at mid-gestation (**C**). The cladogram illustrates that there also wasn’t a distinct phylogenetic enrichment at this point in pregnancy (**D**). Colored circles within cladogram connote taxa at different phylogenetic levels that delineated each reproductive phases (from species to kingdom, innermost to outer rings, respectively). LEfSe indicated that differences in abundance were evident primarily among the cycling and nursing females. Only two bacterial families attained an LDA > 2 in pregnant females (both *Firmicutes*: *Ruminococcaceae* and *Erysipelotrichaceae*).

**Figure 4 microorganisms-11-01481-f004:**
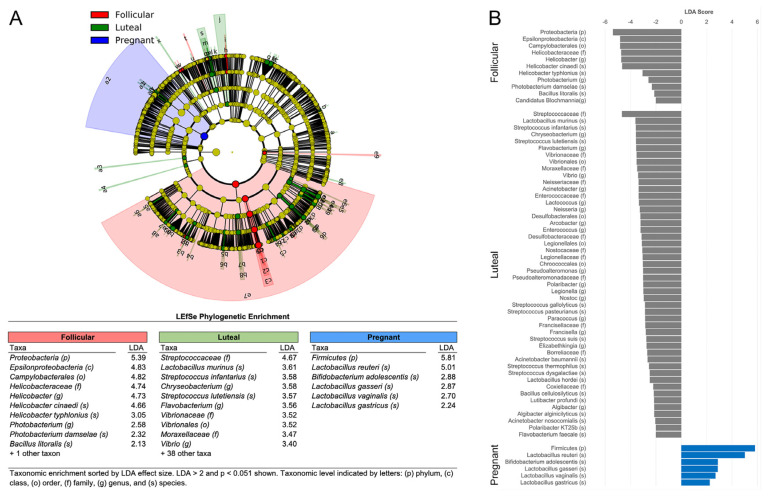
Metagenomic sequencing of fecal specimens from late gestation monkeys identified significant shifts in the bacterial composition of the hindgut when compared to females in the follicular and luteal phases of the menstrual cycle. The height of the blue vector in the cladogram (**A**) reflects the significant enrichment of Firmicutes, including 4 species of *Lactobacillus*, and *B. adolescentis*, and the right and green vectors represent the follicular and luteal phases, respectively. The graphic plot of LDA scores (**B**) illustrates more abundant taxa during the follicular and luteal phases of the menstrual cycle with gray bars, and the blue bars reflect taxa more prominent in the pregnant females.

**Figure 5 microorganisms-11-01481-f005:**
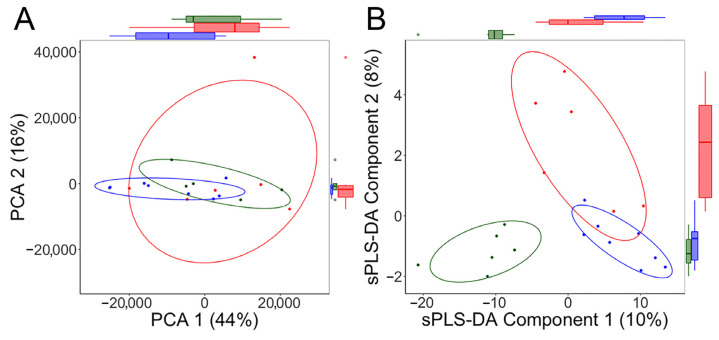
PCA indicated a more homogeneous clustering of gene dispersion in late pregnancy (blue dots and ellipse) (**A**). The overlap conveyed that there was not a global change in the bacterial metagenome in the hindgut of pregnant females. In contrast, when bacterial genes that distinguished each reproductive phases were statistically delineated and considered as a group, the sPLS-DA identified a distinct clustering in the late-pregnant and cycling females (**B**). A 2-component model provided the best fit with 300 genes in Component 1 from both cycling and pregnant females (horizontal axis) and 10 genes specifically from the follicular phase in Component 2 (vertical axis). A complete list of bacterial genes that differed across the 3 reproductive phases, including the sPLS-DA values, is provided in Appendix A. Reproductive phases are color-coded: follicular, red; luteal, green; pregnancy, blue.

**Table 1 microorganisms-11-01481-t001:** Ten most abundant bacterial genera in reproductive tract and fecal specimens from female monkeys in Study 1a.

Vaginal					Rectal				
Family	Genus	Cycling	Pregnant	Nursing	Family	Genus	Cycling	Pregnant	Nursing
Prevotellaceae	Prevotella	13.9%	16.3%	6.1%	Prevotellaceae	Prevotella	12.5%	18.6%	20.6%
Aerococcaceae	Facklamia	0.8%	6.7%	14.5%	Streptococcaceae	Streptococcus	17.2%	13.8%	8.1%
Lactobacillaceae	Lactobacillus	11.7%	8.9%	4.7%	Lactobacillaceae	Lactobacillus	12.9%	14.1%	10.6%
Streptococcaceae	Streptococcus	2.2%	11.6%	5.5%	Helicobacteraceae	Flexispira	3.2%	4.3%	6.2%
Tissierellaceae	Peptoniphilus	9.8%	2.5%	5.4%	Ruminococcaceae	Ruminococcus	0.9%	2.9%	1.1%
Corynebacteriaceae	Corynebacterium	0.2%	1.8%	8.8%	Paraprevotellaceae	CF231	2.2%	1.1%	2.5%
Tissierellaceae	1-68 (clone)	7.0%	3.0%	8.1%	Veillonellaceae	Dialister	1.8%	1.6%	1.8%
Tissierellaceae	Anaerococcus	3.8%	5.3%	8.1%	Ruminococcaceae	Faecalibacterium	0.9%	1.7%	1.8%
Porphyromonadaceae	Porphyromonas	7.7%	0.3%	2.4%	Spirochaetaceae	Treponema	1.3%	1.7%	1.2%
Actinomycetaceae	Actinobaculum	7.1%	0.2%	0.2%	Erysipelotrichaceae	Catenibacterium	0.8%	0.2%	1.2%

**Table 2 microorganisms-11-01481-t002:** Bacterial genes in the hindgut discriminating late gestation from cycling females in Study 2 *.

		sPLS-DA
**Follicular**		
	Two-component system, chemotaxis family, response regulator CheY	0.66
	Precorrin-4/cobalt-precorrin-4 C11 methyltransferase	0.61
	Flagellar biosynthetic protein FliP	0.31
	Iminoacetate synthase	0.20
	Flagellar hook-associated protein 1 FlgK	0.10
	Flagellar basal-body rod modification protein FlgD	0.07
	Flagellar basal-body rod protein FlgG	0.07
	Phospho-N-acetylmuramoyl-pentapeptide-transferase	0.04
	N-carbamoylputrescine amidase	<0.01
	D-tyrosyl-tRNA(Tyr) deacylase	<0.01
	+2 other genes	

**Luteal**		
	1,2-diacylglycerol-3-alpha-glucose alpha-1,2-glucosyltransferase	−0.16
	L-cystine transport system permease protein	−0.15
	Dihydroorotate dehydrogenase electron transfer subunit	−0.14
	Serine/alanine adding enzyme	−0.14
	Acetoin utilization protein AcuB	−0.13
	7-cyano-7-deazaguanine synthase	−0.13
	Histidinol-phosphatase, PHP family	−0.13
	Hypothetical protein	−0.13
	Two-component system, NarL family, response regulator LiaR	−0.13
	Sulfate permease, SulP family	−0.13
	+128 other genes	

**Pregnant**		
	ATP-dependent DNA helicase RecQ	0.17
	Cysteinyl-tRNA synthetase	0.15
	Hypothetical protein	0.13
	Manganese-dependent inorganic pyrophosphatase	0.13
	Pyrroline-5-carboxylate reductase	0.13
	Phosphopantothenoylcysteine decarboxylase/phosphopantothenate cysteine ligase	0.12
	GTP-binding_protein	0.10
	aspartate-ammonia ligase	0.10
	Hydroxymethylpyrimidine/phosphomethylpyrimidine kinase	0.10
	Cell division protein FtsW	0.10
	+150 other genes	

* Top 10 microbial genes distinguishing each reproductive phase are listed.

**Table 3 microorganisms-11-01481-t003:** Mean (S.E.) progesterone in late gestation compared to the levels in the follicular and luteal phases of the menstrual cycle in female monkeys from Study 2.

	Follicular (*n* = 6)	Luteal (*n* = 6)	Pregnant (*n* = 10)	Sig
Blood Progesterone (ng/mL)	0.27 (0.01)	2.34 (1.04)	3.15 (0.64)	0.017
Fecal Progesterone * (ng/g)	0.75 (0.01)	0.16 (.05)	6.77 (2.27)	0.019

* Progesterone concentrations were calibrated with respect to creatinine in the fecal samples. The progesterone in fecal samples and systemic circulation were determined on the same morning as specimen collection for the microbiome analysis.

## Data Availability

Requests for access to datasets should be directed to ccoe@wisc.edu, and should include a scientific rationale and appropriate professional affiliation.

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
