# Peer review of "Significant Microbial Changes Are Evident in the Reproductive Tract of Pregnant Rhesus Monkeys at Mid-Gestation but Their Gut Microbiome Does Not Shift until Late Gestation"

_microorganisms, 2023, doi:10.3390/microorganisms11061481_

Round 1
Reviewer 1 Report
Significant microbial changes are evident in the reproductive tract of pregnant rhesus monkeys at mid-gestation, but their gut microbiome doesn't shift until late gestation.
This manuscript aims to discuss the changes in the microbiota in both the reproductive and gastrointestinal tracts during pregnancy. Overall, it's a well-designed experiment, but revisions will improve the overall scope and status of the article.
Introduction
Line 49-51: How about other pregnancy adverse events and inflammation in addition to preterm birth?
Line 55-57: "Empirical support for this view includes the demonstration that faecal trans-55 plants from pregnant women into germ-free mice resulted in weight gains and a decrease 56 in insulin sensitivity reminiscent of late gestation." Insert reference.
How do you explain the effect of other factors on change in the microbiome, such as medication, hormonal changes, and physical activity during the pregnancy, in addition to diet?
Line 66: Aim/s and hypothesis should be included in the last paragraph.
How would you link changes in the vaginal microbiome and gut in pregnancy?
Material and methods
Consider including a flow diagram of the study design for easy understanding.
Line 102: are the cycling group also pregnant?
Was the food intake measured, including the intake of fruits and vegetables?
Line 113: Does it mean the microbiome was analysed using 16S rRNA sequencing only in three groups?
It is unclear whether bacteria composition was characterised using 16s rRNA sequencing in all three groups.
2..3. does not discuss DNA isolation as the title suggests.
What was the kit used for DNA extraction? Details are needed in the methods.
The methods section should be rewritten for better clarity.
How did you make the functional predictions? The methods section needs to be precise.
Results
Results Figure 1C should come after Figures A and B in the text. What is the test for significance in Figure C?
It's clearer if lefse results and LDA scores are represented in a graph.
There are inconsistencies in font sizes, font names etc, throughout. Please recheck.
Discussion lacks the predicted function and its relation to changes in the vaginal and gut microbiome.
How do you relate the preclinical data to humans, and what are their implications in the clinical outcomes?
The conclusion can be more specific. Lots of information in the conclusion section should be combined in the discussion.
Careful reading and corrections on grammar and spelling mistakes are needed.
Author Response
We appreciate the opportunity to revise our manuscript. We have incorporated the suggestions and constructive criticism of the 3 reviewers. It was helpful to learn that the logic of the experimental design was not clear and that the rationales for each study needed to be expressed in a more transparent manner. We addressed these concerns by adding a new figure that illustrates the experimental design (Figure 1). In addition, we entirely rewrote the Abstract and provided a new paragraph at the end of the Introduction clarifying the research aims. We now have more transition sentences between sections and organized the presentation of results around only 2 studies, rather than 3 studies. Study 1 focuses on mid-gestation, while Study 2 extends the analysis of the gut microbiome to late gestation. We believe that we have been responsive to the issues raised by the 3 reviewers, and the presentation has been improved. The specific changes are detailed below.
Reviewer 1
Introduction
Line 49-51: How about other pregnancy adverse events and inflammation in addition to preterm birth?
We modified this sentence to now include hypertension/preeclampsia as another cause of preterm birth. However, we did not discuss other causes of preterm birth because it was not a clinical endpoint of our own research. A longer discussion might be viewed as a digression in the first paragraph of the Introduction. In addition, we now state more clearly that the pregnant monkeys assessed in this research went on to deliver viable infants that were full-term. It is estimated that bacteria are present in placental tissues in approximately 30% of preterm births, so it seems reasonable to write that investigating commensal bacteria and dysbiosis is important, because they are important factors contributing to prematurity.
Line 55-57: "Empirical support for this view includes the demonstration that faecal transplants from pregnant women into germ-free mice resulted in weight gains and a decrease in insulin sensitivity reminiscent of late gestation." Insert reference
The reference is now provided at the end this sentence, whereas it had previously been cited prior to the sentence. It is citation #14.
How do you explain the effect of other factors on change in the microbiome, such as medication, hormonal changes, and physical activity during the pregnancy, in addition to diet?
The text has now been modified to convey that clinical studies on women indicated that obesity and gestational weight gain contribute to pregnancy-related changes in gut bacteria, which then motivated many basic science researchers to investigate the effects of diet. With respect to the possible influence of medications, we now state in the Methods section that all monkeys were healthy, and none were being administered medications, especially not antibiotics. Regarding the influence of hormones, our interest in assessing progesterone levels in late gestation was based on a published paper demonstrating that progesterone may affect Bifidobacteria levels in late pregnancy. The rationale for assessing progesterone in late pregnancy in our own research is now stated more clearly at the end of the Introduction. Physical activity would be difficult to investigate in monkeys in a laboratory setting. Because it was not evaluated in our research, it doesn’t seem appropriate to discuss the influence of exercise on the gut microbiome, and might make the Introduction seem unfocused.
Line 66: Aim/s and hypothesis should be included in the last paragraph.
We have now written a new paragraph at the end of the Introduction that more clearly explains the experimental design and research aims.
How would you link changes in the vaginal microbiome and gut in pregnancy?
Information on this point is presented in the Results (see Section 3.4) and it is also considered in the Discussion section. In brief, the increased bacterial diversity in the monkey reproductive tract during pregnancy made the vaginal profiles more similar to the intestinal profile. In Discussion, we also reference studies that found the location of the anus above the vaginal opening in a quadrupedal animal facilitates the transfer of fecal bacteria to the reproductive tract. The presence of Prevotella in the vaginal specimens from monkeys also reflects the transfer and exposure to bacteria from the digestive tract. We briefly discuss how the vaginal microbiome of monkeys superficially resembles the gynecological condition of bacterial vaginosis, but that it is well-tolerated in monkeys and not associated with infertility or pregnancy complications.
Material and methods
Consider including a flow diagram of the study design for easy understanding.
In response to this request, a new figure is now included that illustrates the experimental design. Transition sentences have been written to clarify the connections between experiments. Supplemental Table 1 details the number of monkeys in each experiment, as well as in each reproductive phase. Based on the request of Reviewer 2 for more integration, the research plan is now described as consisting of only 2 studies, rather than 3 studies. We followed the request to streamline this aspect of the presentation.
Line 102: are the cycling group also pregnant?
We have now clarified that cycling refers to female monkeys that were having regular menstrual cycles, and thus weren't pregnant. For further clarity, we initially refer to this reproductive phase as non-pregnant, cycling females. In addition, in Supplemental Materials-Methods, we also report that the timing of menses was monitored to track their menstrual cycles. This assessment enabled us to assess cycling females in both the follicular and luteal phases of the menstrual cycle in Study 2.
Was the food intake measured, including the intake of fruits and vegetables?
We now provide the amount of the biscuits and fruit given to each monkey in grams. This information is stated at the beginning of the Methods section. The primary diet was the standardized biscuits given daily. Fruit and vegetables were provided as an enrichment supplement, which was given for variety, but it was only the equivalent of ¼ apple per monkey, provided 3-4 times per week. Because all monkeys in each experiment were assessed contemporaneously, they would have been given the same fruit on the day before the bacterial assessments. All vaginal and fecal swabs for our research projects were obtained in the early morning and would have been collected before any fruit was distributed on that day.
Line 113: Does it mean the microbiome was analysed using 16S rRNA sequencing only in three groups?
We have now clarified that the bacteria present in all specimens were analyzed with 16S rRNA gene amplicon sequencing. In addition, for the replication of Study 1a (which is now labelled as Study 1b), a second aliquot of the extracted bacterial DNA was used for the metagenomic sequencing. In addition, for the follow-up study evaluating the pregnant females in late gestation, we conducted both 16S rRNA gene amplicon and metagenomic sequencing on the fecal specimens.
2.3. does not discuss DNA isolation as the title suggests. What was the kit used for DNA extraction? Details are needed in the methods. The methods section should be rewritten for better clarity.
We have now rewritten this section. We describe the DNA extraction methods for all experiments in the same section. This change was also made in response to the request by Reviewer 2 to present the research protocols in a more integrated manner.
How did you make the functional predictions? The methods section needs to be precise.
We now provide more details on the KEGG analyses and clarify they were conducted only after significant differences in bacterial composition had been detected. Specifically, gene function analyses were conducted for the vaginal specimens obtained in Study 1a, and based on the the gut bacteria sequencing in late pregnancy (Study 2). Tables are provided in Supplemental Materials-Results that detail the genes and provide summaries of the inferred functional pathways (for example, see Suppl. Table 3). Suppl Table 7 summarizes the sPLS-DA effect sizes for 310 genes that distinguished the gut microbiome of late pregnant females from the nonpregnant, cycling females.
Results
Figure 1C should come after Figures A and B in the text. What is the test for significance in Figure C?
Because the graphs showing beta diversity from the weighted and unweighted analyses required two different panels, we placed the graph of alpha diversity in panel C. Given the concern about discussing alpha diversity first, we have now shifted the presentation of the results on alpha diversity results after the findings on beta diversity. It is an issue only for the first figure illustrating the analysis of vaginal specimens (now labelled as Figure 2). There weren’t any significant differences in alpha diversity or beta diversity in the gut.
In response to the request from Reviewer 3, there is now a separate section on the Statistical Analyses. Information on how differences in alpha diversity were tested is provided. Differences in alpha diversity were analyzed with Kruskal-Wallis tests. The values for the Kruskal-Wallis tests conducted on the 4 alpha diversity indices are reported in Supplemental Materials (Supplemental Table 8). We provided these summary statistics regardless of whether the differences reached statistical significance.
It's clearer if lefse results and LDA scores are represented in a graph.
In response to this request, we now provide a graphic representation of the LDA scores from the second experiment on gut bacteria profiles in late pregnancy (Study 2). We agree that a visual rendering can have more dramatic impact.
However, when many taxa attain statistically significant LDA scores, it is difficult to use a font size that is large enough to be legible. That is especially the case if the graph is included as one panel in a large, multi-panel figure. Therefore, we have continued to report the LDA scores in text and in tabular form for Study 1 and graphed only the LDA scores for the metagenomic sequencing in Study 2. Figure 3 has only 2 panels, and thus the graph could be presented in a larger size (see revised Figure 3). Even in Figure 3, however, we continued to highlight the 6 taxa that were enriched in late pregnancy as text in a small table below the cladogram. In addition, we also included a larger copy of the 3B panel with the LDA scores in a larger, full-page format in Supplemental Materials-Results (Suppl. Figure 4). The larger size permitted us to use a font size for the bacterial taxa that is more legible.
There are inconsistencies in font sizes, font names etc, throughout. Please recheck.
We apologize for the variation in font style and sizes in the preprint version that was sent out for the review. We were unaware that the publisher would not be typesetting the tables and had expected our entire submission would be converted to a single font. Thus, while most of the text in the preprint was changed to Palatino Linotype, which is the primary font used by this journal, other parts, including tables and figure legends were still in the original font used in our submitted manuscript. For the revision, we have modified the tables to the journal’s font, which is Palatino Linotype. The different font sizes evident in some tables reflects the fact that this journal publishes tables created in Excel as picture images. In addition, some tables in Supplemental Materials have so many rows and columns that they cannot be readily reproduced in the same font size as the manuscript text without compromising the table formatting. For example, Supplemental Table 7 has 310 rows. We spread the table across many pages. Within the main body of the manuscript, all text and the tables should now appear in the same font.
Discussion lacks the predicted function and its relation to changes in the vaginal and gut microbiome. How do you relate the preclinical data to humans, and what are their implications in the clinical outcomes?
Many of these points are considered in the Discussion section. For example, we reported that the bacterial diversity in the lower reproductive tract of monkeys resembles the gynecological condition of bacterial vaginosis. In part, this species difference in bacterial diversity can be attributed to greater exposure and transfer of fecal material in a quadrupedal animal. We cited this explanation to a publication on the microbiome of domesticated farm animals. We also highlighted that two bacterial taxa present in the reproductive tract monkeys-- Facklamia and Streptococcus—can be pathogenic in humans, even though they seem to be tolerated in pregnant monkeys and do not have adverse clinical consequences in nonhuman primates. We also highlighted that there were no premature births among the pregnant monkeys evaluated in our studies.
Importantly we found that the shift in the gut microbiome was delayed until late pregnancy in monkeys, and not overtly evident at mid-gestation. The delay concurs with a subset of the published studies on pregnant women, including the ones that report larger bacterial changes in the third trimester.
One of the more striking findings in Study 2 that focused on late gestation was the discovery that progesterone levels are so low in pregnant rhesus monkeys, even in the final weeks before delivery. The blood levels of progesterone were only 2% of the circulating progesterone levels in women during the third trimester. Thus, it may not be surprising that we did not see strong associations with most bacterial taxa. Nevertheless, it is noteworthy that we did find a significant association between the levels of progesterone present in the fecal specimens and the relative abundance of the Family Bifdobacteriacaeae. That association would be in keeping with a published report demonstrating a linkage between progesterone and Bifidobacteria in pregnant women. This association could also be induced in gravid mice by administering progesterone. We also showed the value of considering the amount of progesterone present being excreted via the gut, which would be more proximal to the commensal bacteria.
We tried to provide a more compelling summary of the major findings in the revision. Because the Discussion is already long, rather than add more topics, we included a new citation to a recent paper that provides an overview of the relevance of monkey models to clinical and gynecological research and practice.
Hugon, A.M., Golos, T.G. Nonhuman primate models for understanding the impact of the microbiome on pregnancy and the female reproductive tract. Bio Reprod 2023, 1-16. https://doi.org/10.1093/biolre/ioad042
Careful reading and corrections on grammar and spelling mistakes are needed.
Thank you for alerting us to the need to be more conscientious in proofing for typos and problematic sentence structures. We checked the manuscript for spelling errors. Please note that we used the American English form of common words, rather than British spelling.

Reviewer 2 Report
My main issue with this manuscript is that it is incredibly confusing and in my opinion, disorganized. The methods is all over the place and the authors seem to be doing three separate experiments with no clear reason why it is not linked into one large investigation. It makes understanding the conclusions difficult. Also, the term sMGS is not used (to my knowledge) in common microbiome papers and it is not clear why the authors did both this AND 16S. The abstract does not mention the progesterone analyses and makes no mention of the '3 experiments' that are discussed. The methods are not organized chronologically but more based on these 3 projects which makes understanding what/why the authors did what they did incredibly muddled. The figures seem to be presented exactly as the methods spit them out with no rhyme or reason in their organization. The sMGS analyses are not clear nor are the PCA plots. Table 2: what is DA and what are these values?
This paper needs a massive overhaul in order to make sense as a publishable document. I suggest major revisions and advise the authors to try to present the three experiments as ONE in a logical manner.
Author Response
…. the authors seem to be doing three separate experiments with no clear reason why it is not linked into one large investigation.
In response to this concern about the logic and rationales for the experimental plan not being clearly presented, we made several changes. We merged the prior Study 1 and Study 2 into a single study with 2 phases. They are now labelled as Study 1a and 1b. The purpose of the follow-up experiment was primarily to replicate the 16S rRNA gene amplicon results on the gut microbiome from the first experiment. The conclusion that there wasn’t a significant shift in the gut microbiome at mid-gestation was confirmed. Therefore, it does make sense to consider the replication as a second phase of Study 1. The conclusion about stability of the bacterial composition at mid-gestation was also affirmed when we analyzed a second aliquot of the extracted DNA with metagenomic sequencing. Those results laid the groundwork for then shifting the timing of fecal sample collection later in pregnancy closer to term.
This experimental plan is now illustrated in Figure 1. Reviewer 1 had recommended that we include an illustration of the experimental plan. We also wrote a new paragraph at the end of the Introduction, which details the research plan and the aims of each study. In brief, Study 1 is an assessment of the vaginal and gut microbiome at mid-gestation, while Study 2 is an assessment of the gut microbiome in late pregnancy. In addition, we now provide more transition sentences to provide better bridging between different sections.
The Abstract was entirely rewritten to present the research plan and findings in a more integrated manner. However, we should mention that the journal guidelines limit the Abstract to only 200 words. Therefore, we could not include all of the important details in the Abstract. We have also provided more information in Supplemental Materials. For example, Supplemental Table 1 shows the specific number of monkeys in each experiment and in each reproductive phase. We have clarified in several locations that Study 1 evaluated the vaginal and gut microbiome at mid-gestation, whereas Study 2 focused only on the gut microbiome in late pregnancy. This difference reflected the fact that we had already identified large changes in the vaginal microbiome by mid-gestation, but the shift in the gut microbiome was not manifest until late gestation.
We purposefully refrained from publishing the results on the gut microbiome from Study 1a until they were confirmed in Study 1b. We then felt it was critical to be able to put the null findings for the mid-gestation gut microbiome in the correct context, which was the reason for determining in Study 2 if pregnancy-related changes in the gut bacteria occurred later closer to term.
Also, the term sMGS is not used (to my knowledge) in common microbiome papers.
We agree that metagenomic sequencing is not typically abbreviated in most articles. However, there is some precedent in the literature for using abbreviations such as MGS, sMGS or tMGS. But given the concern of Reviewer 2, we have now written it out in words and not used an abbreviation or acronym.
Below are two examples of published papers that did use abbreviations for metagenomic sequencing:
Flurin L et al. Clinical use of a 16S ribosomal RNA gene-based Sanger and/or next generation sequencing assay to test preoperative synovial fluid for periprosthetic joint infection diagnosis. Mbio 2022. 13, 6.
Rodino K. et al. Retrospective review of clinical utility of shotgun metagenomic sequencing testing of cerebrospinal fluid from a US tertiary care medical center. J Clin Microbio 2020, 58(12), e01729-20
However, in deference to this concern, we metagenomic sequencing is always written out in words.
The sMGS analyses are not clear nor are the PCA plots. Table 2: what is DA and what are these values?
In response to this request and a related one from Reviewer 3, we now have a separate section describing the data analytic strategy at the end of the Methods section. We also provide summaries of the statistical test values in Supplemental Materials-Results (Supplemental Table 8). We apologize that the meaning of the abbreviation for ‘discriminant analysis’ was unclear, and now identify it more fully as referring to the values from the sPLS-DA tests.

Reviewer 3 Report
The authors of this manuscript evaluated changes in the bacterial profiles of both the reproductive and gastrointestinal systems during pregnancy, by comparing these profiles between cycling and lactating females in three experiments. I have the following two comments:
1. Abstract: The phrase “The stability of community structure and composition in the gut at mid-gestation was then replicated with additional monkeys” should be rephrased. The community structure and composition refers to what?
2. The authors should also include a “Statistical analysis” subsection at the end of the Methods.
Author Response
Abstract: The phrase “The stability of community structure and composition in the gut at mid-gestation was then replicated with additional monkeys” should be rephrased. The community structure and composition refers to what?
We have now clarified that this sentence referred to bacterial composition and community structure. The Abstract has also been entirely rewritten in response to the request of Reviewer 2 for a more integrated presentation of the research plan and results. Pregnancy-related changes in the gut microbiome were manifest more by changes in bacterial composition rather than by significant shifts in community structure. However, the exception was that significant changes in both alpha and beta diversity were evinced in the lower reproductive tract by mid-gestation.
- The authors should also include a “Statistical analysis” subsection at the end of the Methods.
We apologize for not including this section previously, and describing the statistical tests only as each measure was presented. There is now a separate section on the statistical testing and data analytical strategy.
We hope the reviewers feel we have been responsive to their concerns and have responded in good faith to the requests. The changes have improved the presentation. Hopefully the research plan and the rationales for sequential experiments are more clearly articulated. The purpose of Study 1b was to replicate our initial findings on the gut microbiome at mid-gestation. When it led to the same conclusion, we then shifted the assessment of the gut microbiome to later in pregnancy closer to term. The latter time point for sample collection yielded the more expected finding of a significant change in the bacterial composition in the hindgut.
We appreciate that additional time and work is required for a re-review. We hope the reviewers and Guest Editor conclude we have been responsive to the requests and concerns.

Round 2
Reviewer 1 Report
Thanks for addressing the comments. I am happy with the revisions.
Spelling and grammar should be checked.
Author Response
Thank you for alerting us to the fact that our submission would benefit from another round of proofing. We carefully checked for spelling errors, typos and grammatical issues. In several instances, we also divided long sentences into two shorter sentences to facilitate the ease of reading.
We also carefully reviewed the microbial terminology and clarified and/or corrected several terms. For example, we now use Bifidobacterium when referring to a single taxon.
This additional scrutiny allowed us to catch several small formatting and spacing errors in the preprint at a stage when they could still be easily corrected. We apologize that we didn’t catch them before the preprint was sent for the re-review, but we first saw this version of the preprint at the same time as the reviewers.
The request for additional proofreading has enabled us to significantly improve our submission. We appreciate your help in urging us to proofread more closely.
Reviewer 2 Report
I have no major issues with the current version of the paper, my comments were addressed, as were those of the other reviewers. I request that the authors make their data publicly available on the SRA database, which is standard procedure for 16S and metagenomic data. Additionally, the authors should provide a github with all the qiime2 and R code used in the project.
Author Response
We are in total agreement about the importance of transparency in science. We also appreciate that this new information about the microbiome in the reproductive and digestive tracts of monkeys may have other potential applications beyond the pregnancy-related analyses we conducted. We can prepare raw data files to deposit and share them in the public domain, such as at https://www.ncbi.nlm.nih.gov/sra/docs/submit/. We can also post the Qiime2 and R codes used in our analyses to one of our GitHub accounts.
One of the senior authors (CLC) has extensive prior experience with sharing data in a public portal because he directed the Biomarker Core for a large study of health and aging in older Americans for over a decade (https://midus.wisc.edu/). All biological and social data from this large project were shared publicly. This sharing and dissemination resulted in over 1000 publications written by scientists from around the world.
Thank you for encouraging us to ensure there will be maximal use and gain from the unique dataset we acquired. We agree that this amount of information on 72 monkeys is not easy to obtain. Our breeding program is particularly unique in the use of standardized husbandry practices, which allowed us to obtain synchronized specimens from non-pregnant, pregnant, and lactating females at the same time so that they all could be sequenced as a single batch.